# Study on Laser Polishing of Ti_6_Al_4_V Fabricated by Selective Laser Melting

**DOI:** 10.3390/mi15030336

**Published:** 2024-02-28

**Authors:** Shuo Huang, Junyong Zeng, Wenqi Wang, Zhenyu Zhao

**Affiliations:** 1School of Mathematics and Information Engineering, Xinyang Institute of Vocational Technology, Xinyang 464000, China; h6955013@163.com; 2School of Sino-German Robitics, Shenzhen Institute of Information Technology, Shenzhen 518172, China; zjy2070292047@163.com (J.Z.); 18312215870@163.com (W.W.)

**Keywords:** Ti_6_Al_4_V, laser polishing, surface roughness, microstructure, wear, dry fraction

## Abstract

Laser-based additive manufacturing has garnered significant attention in recent years as a promising 3D-printing method for fabricating metallic components. However, the surface roughness of additive manufactured components has been considered a challenge to achieving high performance. At present, the average surface roughness (Sa) of AM parts can reach high levels, greater than 50 μm, and a maximum distance between the high peaks and the low valleys of more than 300 μm, which requires post machining. Therefore, laser polishing is increasingly being utilized as a method of surface treatment for metal alloys, wherein the rapid remelting and resolidification during the process significantly alter both the surface quality and subsurface material properties. In this paper, the surface roughness, microstructures, microhardness, and wear resistance of the as-received, continuous wave laser polishing (CWLP), and pulsed laser polishing (PLP) processed samples were investigated systematically. The results revealed that the surface roughness (Sa) of the as-received sample was 6.29 μm, which was reduced to 0.94 μm and 0.84 μm by CWLP and PLP processing, respectively. It was also found that a hardened layer, about 200 μm, was produced on the Ti_6_Al_4_V alloy surface after laser polishing, which can improve the mechanical properties of the component. The microhardness of the laser-polished samples was increased to about 482 HV with an improvement of about 25.2% compared with the as-received Ti_6_Al_4_V alloy. Moreover, the coefficient of friction (COF) was slightly reduced by both CWLP and LPL processing, and the wear rate of the surface layer was improved to 0.790 mm^3^/(N∙m) and 0.714 mm^3^/(N∙m), respectively, under dry fraction conditions.

## 1. Introduction

As a lightweight metal material, Ti_6_Al_4_V alloy has been widely used in biomedical, aerospace, automobile, ship, petrochemical, and other fields owing to its advantages of high strength, good corrosion resistance, good bio-compatibility, and high specific strength [1,2,3,4]. Nevertheless, the traditional fabrication and manufacturing of titanium alloy products can lead to massive material wastage and high production times [5]. With the advancement of additive manufacturing technology, an increasing number of researchers are devoting their attention to the utilization of selective laser melting (SLM) technology for the fabrication and processing of titanium alloy products [6]. The SLM technology utilizes a high-power laser to selectively melt the powder in accordance with the design methodology, enabling layer-by-layer printing of the final product [7,8,9]. The SLM technologies present a viable solution for the rapid production of intricate freeform components without the need for traditional tooling [10]. However, SLM technology still has considerable limitations compared with traditional and subtractive manufacturing processes [11]. The dimensional accuracy and surface quality of the parts produced through laser additive manufacturing are comparatively lower than those achieved by conventional machining methods, thereby impeding the widespread adoption of this innovative technique [12,13,14]. Additionally, the presence of subsurface porosity in LPBF parts also limits the standardization and widespread nature of the technology due to the deterioration of the mechanical properties, especially fatigue life [15].

To cope with these issues, laser polishing is used as one of the newest thermal energy-based method to attain highly superior surface finish levels [16]. Laser polishing for metals is typically based on melting a thin surface layer of the workpiece followed by surface smoothening using interfacial tension. In this case, the existing material of the “hills” is used to fill the “valleys” in the surface texture [17,18,19]. Ma et al. [20] used a fiber laser to polish the additive manufactured (AM) Ti_6_Al_4_V alloys parts. The findings demonstrated a reduction in surface roughness of the Ti_6_Al_4_V alloys from 5 μm to 1 μm. Moreover, the wear resistance and micro-hardness of the laser-polished surfaces were better than the as-received surfaces. Marimuthu et al. [21] examined the effects of laser polishing parameters on surface topology and roughness. By controlling the laser power, speed, and beam offset, a significant reduction in surface roughness was achieved, with the initial Ra value of approximately 10.2 μm being reduced to 2.44 μm. Shen et al. [22] applied a two-step surface treatment combined with a picosecond laser polishing and pulsed laser polishing to enhance the surface quality of the Ti_6_Al_4_V fabricated by laser metal deposition. The result showed that the surface roughness was reduced by about 90% after the two-step surface treatment. Additionally, the laser polishing induced the phase transformation resulting in a finer grain size. Xu et al. [23] conducted a laser polishing experiment on the AM TiAl alloy. The results denoted that the CW laser-polished surface boasted better surface quality, wear resistance, and corrosion resistance performance than the initial surface. Liu et al. [24] combined the continuous wave laser polishing and pulsed laser polishing to improve the surface quality and superficial microstructure of the LDED Inconel 718 superalloy, which decreased the surface roughness from 15.75 μm to 0.23 μm.

According to the above, lots of works had been carried out on additive manufacturing alloys’ components’ laser polishing. However, limited research has been conducted on the impact of various laser types on the surface quality of SLM Ti_6_Al_4_V alloy samples. In this paper, a CW laser and a pulsed laser were used to polish the Ti_6_Al_4_V alloy’s components fabricated using selective laser melting (SLM). The surface morphology, microstructures, phase constituent, microhardness, and wear resistance of the laser-polished area were investigated systematically.

## 2. Experimental Details

### 2.1. Experimental Material and Equipment

The Ti_6_Al_4_V alloy samples with size of 100 × 100 × 8 mm^3^ were fabricated by a selective laser-melted (SLM) manufacturing system from Ti_6_Al_4_V alloy powders, which were directly polished by laser polishing after sand blasting. The morphology of Ti_6_Al_4_V alloy powders was shown in Figure 1. Clearly, the powders displayed a characteristic spherical shape with a relatively uniform size distribution.

The laser polishing experiment was conducted in two stages, specifically, continuous wave (CW) laser system and pulsed laser polishing system, as shown in Figure 2. The CW laser polishing system consisted of a continuous wave laser (MFSC-1000, Chuangxin Laser Co., Ltd., Shenzhen, China, and the wavelength λ = 1064 nm, maximum laser power P = 1000 W, spot diameter D = 0.27 mm) and a scanning galvanometer (F20PRO-3-MS, Fretak Laser Technology Co., Ltd., Jiangsu, China) with a focal length of 720 mm. The pulsed laser polishing system included a pulsed laser (YDFLP-CL-300-10-W, Shenzhen JPT Opto-electronics Co., Ltd., Shenzhen, China, and wavelength λ = 1064 nm, laser repetition frequency f = 100 kHz, pulsed duration σ = 240 ns, spot diameter D = 0.34 mm) and a scanning galvanometer (F20PRO-3-MS). The laser energy of a continuous wave laser exhibited a top-hat distribution, whereas that of a pulsed laser followed a Gaussian distribution. To mitigate the interference of impurities on the surface of the samples with the experimental results, the samples were reposed in the ultrasonic cleaner to clean for 15 min. When the laser was applied, a 99% argon was injected into the two-dimensional rotary platform as a protective gas to avoid oxidation reaction. The employed scanning strategy for laser polishing was illustrated in Figure 2b, and the dimensions of the polishing region were approximately 10 × 10 mm^2^. The laser processing parameters are tabulated in Table 1. Laser polishing was performed on the top surface of all the samples and the schematic views of the CW laser polishing and pulsed laser polishing are shown in Figure 2c,d.

### 2.2. Surface Characterization

The surface roughness and topography were observed by Laser Confocal Microscope (Mahr, MarSurf-CM, Göttingen, Germany) on a 2.5 × 2.5 mm^2^ area at three locations, and the average was calculated. The microstructure and elemental content were analyzed with a scanning electron microscope SEM (ZEISS Gemini 300, Gina, Germany), fitted with energy dispersive X-ray spectroscopy (EDS). The phase composition of samples was analyzed with an X-ray fluorescence spectrometer (Bruker, D8 Advance, Billerica, Germany). The hardness was measured with an automatic turret microhardness tester (HV-1000A, Laizhou Huayin Testing Instrument Co., Ltd., Laizhou, China) under a 200 g load for a dwell time of 10 s. The fraction tests were conducted using a multifunctional friction wear tester (GF-1, Lanzhou, China) under ambient conditions. The Si_4_N_3_ ceramic grinding sphere with a diameter of 5 mm was utilized as the friction pair for conducting friction wear tests [25,26]. The applied load was 15 N, accompanied by a rotational velocity of 300 r/min and a reciprocating distance of 5 mm. The total duration of sliding friction wear amounted to 12 min. At least three repeat tests were carried out to check repeatability and a randomly selected set of experimental samples was analyzed.

## 3. Results and Discussions

### 3.1. The Effect of Laser Polishing on Surface Roughness and Surface Topography

Figure 3 shows the surface roughness and the three-dimensional morphology of the as-received, CWLP, and LPL samples observed with the laser scanning confocal microscope. It can be seen that there were un-melted powder particles on the surfaces of the Ti_6_Al_4_V alloy samples following SLM, and the as-received sample exhibited the highest roughness of Sa = 6.29 μm. The PLP-processed surface resulted in the lowest roughness with a Sa value of 0.84 μm, while the CWPL-processed surface demonstrated a median roughness with an Ra value of 0.94 μm, as illustrated in Figure 3b,c. Compared with the initial sample, the average surface roughness of the CWLP and PLP samples were reduced by about 85.1% and 86.6%, respectively. This surface evolution is primarily attributed to the utilization of different laser polishing techniques. Continuous wave laser polishing (CWLP) offers a more consistent output energy, which effectively promotes material melting and facilitates the redistribution of molten pool material. On the other hand, pulsed laser polishing (PLP) provides an instantaneous output energy that greatly aids in vaporizing un-melted particles on the surface of Ti_6_Al_4_V alloys.

### 3.2. The Effect of Laser Polishing on Microstructure Evolution and Phase Composition

The microstructure, as well as the surface elemental distribution of the SLM Ti_6_Al_4_V with and without laser polishing, were observed by SEM. It can be seen from Figure 4a–f that there were many irregular bulge and pit features present on the as-received specimen surface, which increased the average surface roughness. The EDS result revealed that the surface of the Ti_6_Al_4_V sample exhibited an even distribution of Ti, Al, V, O, and C. After CWLP processing, the irregular protrusions and pits on the sample surface were eliminated, revealing distinct laser scanning tracks, as depicted in Figure 4g,h. This phenomenon can be attributed to the redistribution of the molten pool facilitated by both surface tension and gravity, resulting in the formation of a high-quality surface. Moreover, micro-cracks also appeared on the surface of the CWLP samples. This was due to the rapid cooling rate during laser polishing, which causes the remelted layer to solidify faster than the interior. As a result, significant thermal stress is generated in the thick heat-affected layer and a continuous thermal cycle takes place, ultimately leading to crack formation [27]. These cracks will affect the service life of the product due to poor fatigue strength, which will hinder its application in engineering practice. However, step structures appeared on the surface of the PLP samples, as shown in Figure 4m,n. The different orientation of the crystal lattice or the glide planes lead to different deformations of the individual grains. In addition, the size and growth direction of the grains in the molten pool also have an important influence on the formation of the step structure [28]. The results also demonstrated that laser polishing has effectively homogenized the distribution of elements.

Figure 5 shows the elemental content of the initial surface, and the CWLP and LPL sample surfaces. It can be seen that the content of Ti on the surface of the initial sample is 51.99%, accounting for the largest proportion, followed by the content of O accounting for 32.18%, while the content of Al, V, and C is only 9.53%, 2.27% and 4.02% respectively. The preparation process of SLM Ti_6_Al_4_V alloy samples is conducted in an ambient air environment, leading to a significant oxidation reaction between the material and the abundant oxygen present at high temperatures, consequently resulting in a substantial oxygen content on the surface of the initial sample. After CWLP process, the Ti content accounts for the largest percentage increased to 83.2%, while the O content decreased to 4.92%. The primary reason is that the laser processing is conducted within an argon environment, leading to an overflow of O on the sample surface and subsequently resulting in a reduction in surface O content. Additionally, there is a slight decrease observed in both the C content and Al content. The same trend of element contents also appears on the surface of the pulsed laser polished sample.

In order to further study the effect of laser polishing on the remelted layer, the cross-section microstructures of the Ti_6_Al_4_V samples under different laser polishing methods were observed, as shown in Figure 6a,b. The average thickness of the remelted layer in the CWLP and LPL samples was 101 μm and 70 μm, respectively. The remelting zone exhibited a significant presence of dendritic structure and needle-shaped lamellar regions, comprising α′ phase martensite. Due to rapid melting and solidification, complex heat effects occur on the surface of the material resulting in changes in microstructure during the laser polishing process. When the temperature exceeds the transition temperature of the β phase (990 °C), the martensite will completely transform into the β phase martensite [29]. Then, during the cooling period, almost all of the β phase martensite will transform into α′ phase martensite due to the cooling rate of the sample surface exceeding 410 K/s, so that a large amount of α′ phase martensite is found in the remelting zone [30]. Furthermore, the remelting layer of the PLP samples exhibits a higher proportion of α′ phase martensite compared with that of the CWLP samples. In contrast to CWLP that have a steady energy input, pulsed lasers exhibit an intermittent energy input, resulting in a higher cooling rate and an increased formation of martensitic structures on the surface of laser-polished samples. According to the study of Ma et al. [16], in the thermal cycles of pulsed laser processing, the cooling rates of Ti_6_Al_4_V were calculated as 1.209 × 10^4^ K/s, which are much higher than 410 K/s. Due to the rapid solidification and cooling rates, the bcc β phase undergoes a complete transformation into the metastable hcp α′ phase martensite phase through a diffusionless and shear-type transformation process. Therefore, the remelting layer of the PLP sample exhibited a higher proportion of α′ phase martensitic structure.

The phase compositions of the as-received, CWLP, and LPL samples were analyzed using XRD, and the results are presented in Figure 7. The as-received SLM Ti_6_Al_4_V exhibited a distinct diffraction peak corresponding to the α phase and a less pronounced diffraction peak associated with the β phase, indicating that the initial Ti_6_Al_4_V alloy sample primarily consists of dual-phase martensite composed of α-Ti and β-Ti phases. The α phase martensite mainly grew along the crystallographic plane of (100), (002), (101), (102), (103), (112), (201), while the β phase martensite mainly grew along the crystallographic plane of (110). After the laser polishing process, only the strong diffraction peaks in α′-Ti were observed and the weak diffraction peaks in the β-Ti phase disappeared. This is attributed to the martensite transformation from the β phase, which retains a metastable or stable condition after cooling to the α′ phase due to the nucleation and growth to a Windmanstatten pattern of the α′ phase during the cooling process [31]. This experiment result is consistent with the conclusion that was drawn by Ma et al. [20] in exploring the effect of laser polishing on the microstructure of the Ti alloy. Due to its identical hexagonal close-packed structure, it exhibits an indistinguishable peak position in the XRD spectrum, rendering it unsuitable for accurately discerning different phases. Notably, a slight high-angle shift is observed in the (110) diffraction peak of the α′ phase following laser polishing, indicating the presence of residual compressive stress. The application of a laser heat source will induce localized thermal effects on the surface of the polished sample, leading to residual stress upon solidification.

### 3.3. The Effect of Laser Polishing on Microhardness

Figure 8 displays the microhardness depth profiles of laser-polished specimens in argon environments. The microhardness of the as-received sample exhibited slight fluctuations with increasing depth, ultimately stabilizing at 385 ± 10.2 HV. The microhardness of the top layer in the CWLP and LPL samples increased to 482 ± 5.6 HV, indicating about a 25.2% improvement in hardness, which is consistent with the conclusions drawn by Zhou et al. [32] when they investigated the effect of laser polishing on the micro-hardness and surface characteristics of Ti_6_Al_4_V alloy parts. This phenomenon can be attributed to the formation of martensitic phases, which is caused by a high density of dislocations and numerous phase boundaries resulting from the twin needle-like structure. Therefore, the hardness at the remelting layer is higher than that of the base metal. It can be found that the depth of the affected layer measured approximately 200 μm. At the same depth, the hardness of the PLP samples was slightly higher than that of the CWLP samples, which is consistent with the results of the cross-section microstructure (see Figure 6).

### 3.4. The Effect of Laser Polishing on Surface Tribology Properties

Figure 9 shows the 3D morphology and line cross-sectional profile of the wear track of the as-received, CWLP, and LPL sample surfaces under dry friction conditions. It can be found that the wear track depths of the as-received, CWLP, and LPL samples were measured to be approximately 120 μm, 114 μm, and 101 μm, respectively. Similarly, their widths were found to be 1678 μm, 1592 μm, and 1585 μm. These findings suggest that the wear volume on the laser-polished sample surfaces were comparatively smaller than those on the initial surface.

Figure 10 illustrates the COF and wear rates of as-received, CWLP, and PLP samples under dry friction conditions with a normal load of 15 N, a motor speed of 300 r/min, and a sliding distance of 5 mm. It can be seen from Figure 10a that the as-received sample exhibited an initial COF of 0.350 during the wear test, which then gradually increased and eventually stabilized at 0.643. The COF of the CWLP sample showed slight fluctuations and reached a steady-state value of 0.570, indicating a similar change trend to that observed in the initial sample. However, the PLP sample had an initial COF of 0.340, which gradually increased to 0.476 after 12 min and subsequently decreased to 0.436. It is evident that the PLP sample produced the lowest COF, followed by the CWLP sample, while the as-received sample exhibited the highest COF. The average COF of the as-received, CWLP, and PLP samples were 0.570, 0.509, and 0.434, respectively. Compared with the initial sample, the average friction coefficient of the CWLP and PLP sample surfaces were reduced by about 10.7% and 23.9%, respectively. This phenomenon may be related to the effects of surface roughness and microhardness. Amanov et al. [33] proposed that lower roughness leads to lower COF, while an increase in hardness is responsible for the enhanced wear resistance. Compared with the as-received and CWLP samples, the PLP samples exhibited the lowest surface roughness and the highest micro-hardness (see Figure 3 and Figure 8), resulting in a reduced friction coefficient. In addition, the COF of the Ti_6_Al_4_V sample is also related to sliding speed, wear time, applied load force, wear material, and environment [34,35,36].

Specific wear rate is an important parameter to characterize the wear resistance of the sample surfaces; it can be defined as follows [37]:(1)$$W=\frac{\Delta V}{L*D}$$
where ΔV is the loss volume of the wear, L is the applied load in the experiment, and D is the sliding distance. The special wear rates of the as-received, CWLP, and PLP sample surfaces were displayed in Figure 10b. The wear rate of the CWLP and LPL samples was found to be 0.790 mm^3^/N·m and 0.714 mm^3^/N·m, respectively, exhibiting an improvement of approximately 6.4% and 15.9% compared with the as-received sample. The re-molten layer after CWLP and LPL processing had a depth of approximately 200 μm, containing a significant amount of hard α′ phase martensite. As a result, the wear resistance of the polished samples was significantly enhanced compared with that of the substrate. However, there were some cracks on the CWLP sample surface, which reduced the wear resistance of the surface. Previous studies [38] have primarily employed surface-treatment methods to enhance the wear resistance of the Ti_6_Al_4_V alloy by modifying its microstructure, phase composition, and residual stress state.

To further investigate the impact of laser polishing on the wear behavior between the SLM Ti_6_Al_4_V sample and the Si_3_N_4_ ceramics, SEM and EDS were employed to analyze the surface morphology and chemical composition of the wear track of the as-received, CWLP, and LPL samples under dry sliding conditions, as depicted in Figure 11. It can be observed that the wear track exhibits numerous grooves and scratches aligned parallel to the sliding direction, along with a significant presence of fine wear debris, pronounced delamination, and crack. This demonstrates the occurrence of plastic deformation in certain regions during friction. The phenomena indicate that the abrasive wear are the main wear mechanisms [39]. The wear track surfaces of the CWLP and LPL samples were illustrated in Figure 11b,c, respectively. Similar characteristics of wear grooves along with a significant amount of wear debris were also observed on the wear tracks. Furthermore, the width of the wear tracks for both the CWLP and LPL samples was narrower compared with that of the initial sample, which aligns with the findings presented in Figure 9. The contents of O on the surface of the laser-polished samples were slightly increased after undergoing the friction and wear test, indicating the occurrence of a certain oxidation reaction during this process. The aforementioned findings collectively demonstrate a significant enhancement in the wear resistance of TiAl alloy surfaces following laser polishing, as compared with their initial state.

## 4. Conclusions

In this research, the effect of CWLP and LPL processes on the surface morphology, microstructure, hardness, and wear resistance of SLM Ti_6_Al_4_V alloy parts were investigated in detail. The obtained findings can be summarized as follows:(1)The surface roughness of the as-received sample can be reduced from 6.29 μm to 0.84 μm through the PLP process, achieving a remarkable reduction rate of 87%. In comparison, the CWLP process resulted in a slightly higher surface roughness of 0.94 μm. However, cracks were generated on the surface of samples treated with the CWLP process.(2)The microstructure of the SLM Ti_6_Al_4_V alloy after laser polishing consisted of a dendritic structure and needle-shaped lamellar regions, comprising α and α′ phase martensite. The average thickness of the remelted layer in the CWLP and LPL samples was 101 μm and 70 μm, respectively.(3)The microhardness of the laser-polished sample surface exhibited a significant increase and stabilized at about 482 HV, representing an approximately 25.2% improvement compared with the as-received sample, which is attributed to the increased presence of the α′ phase in the remelting layer. Additionally, the affected layer’s depth measured approximately 200 μm.(4)The laser polishing process effectively enhanced the tribological performance of SLM Ti_6_Al_4_V when sliding against a Si_3_N_4_ ceramics ball under dry conditions. The PLP sample exhibited the lowest average COF (0.434) and wear rate (0.714 mm^3^/N·m), followed by the CWLP sample, while the as-received sample owned the highest average COF (0.590) and wear rate (0.844 mm^3^/N·m).

## Figures and Tables

**Figure 1 micromachines-15-00336-f001:**
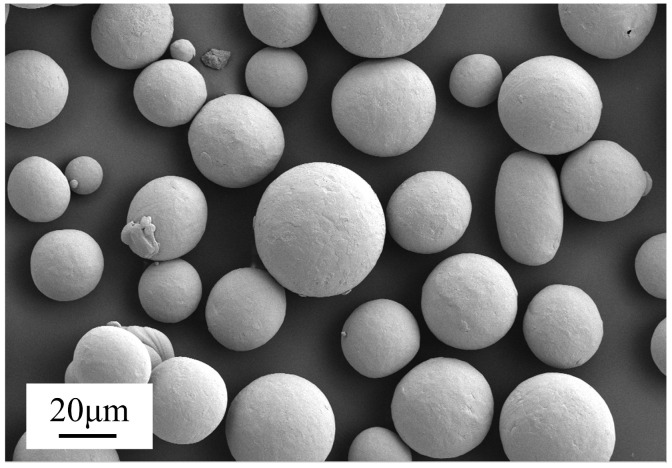
SEM of Ti_6_Al_4_V alloy powders.

**Figure 2 micromachines-15-00336-f002:**
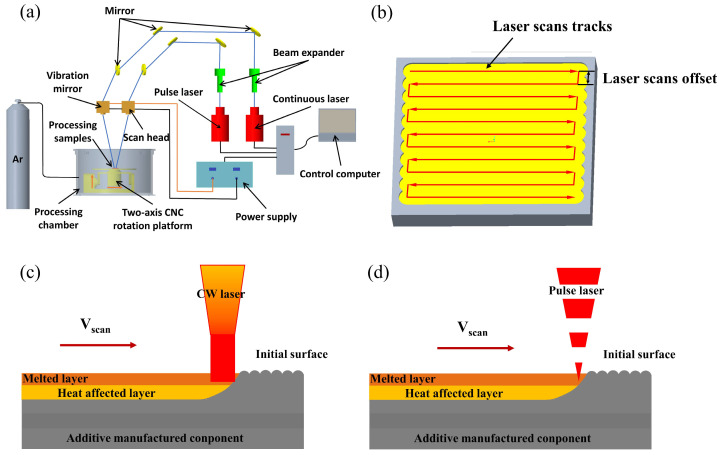
(**a**) Schematic diagram of experimental setup, (**b**) the scanning strategy of laser polishing, (**c**) schematic view of CW laser polishing process, (**d**) schematic view of pulsed laser polishing process.

**Figure 3 micromachines-15-00336-f003:**
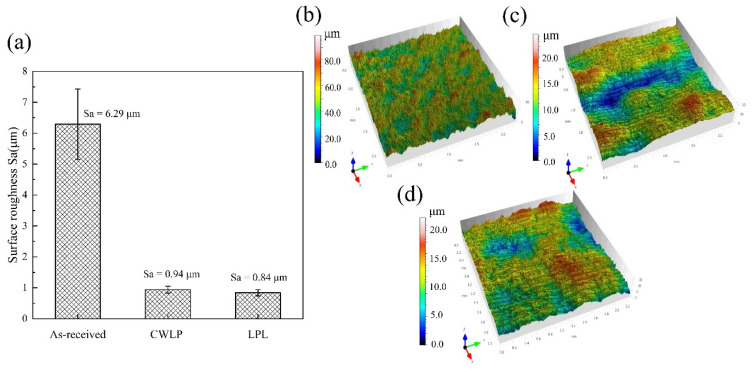
The surface roughness and the three-dimensional morphology: (**a**) Surface roughness, (**b**) 3D morphology of the as-received sample, (**c**) 3D morphology of the CWLP sample, (**d**) 3D morphology of the LPL sample.

**Figure 4 micromachines-15-00336-f004:**
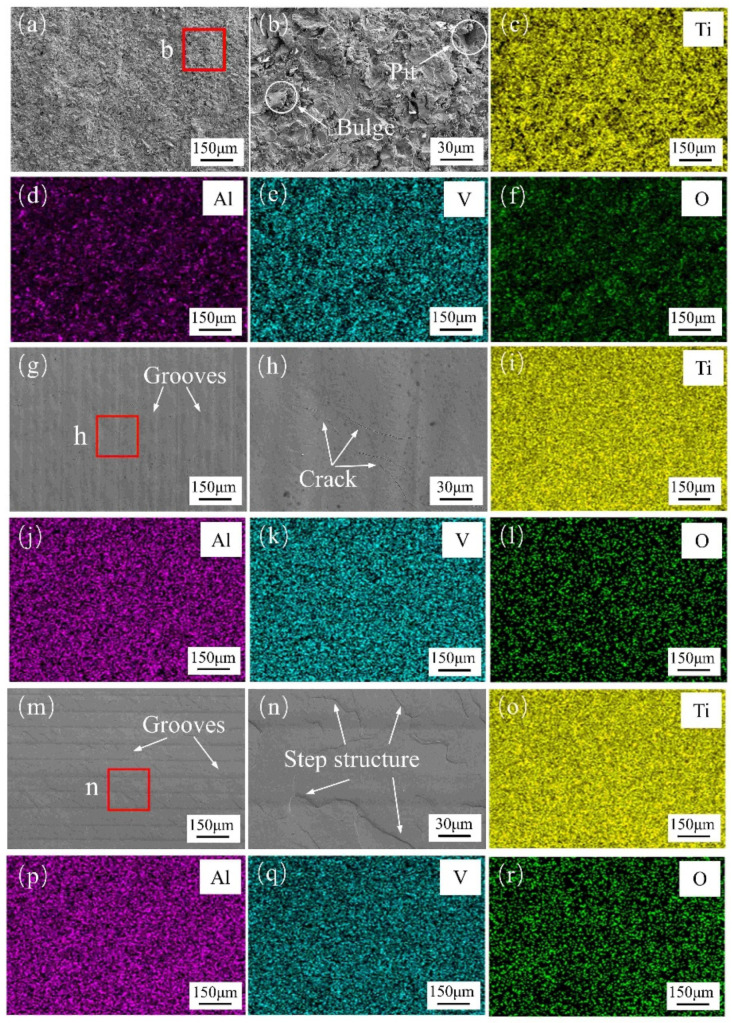
Microstructure in sample surface and EDS elemental distribution of SLM Ti_6_Al_4_V with and without laser polishing: (**a**) Microstructure of top layer of as-received sample, (**b**) High resolution map, (**c**–**f**) Distribution of Ti, Al, V, and O in the as-received sample, (**g**) Microstructure top layer of the CWLP sample, (**h**) High resolution map, (**i**–**l**) Distribution of Ti, Al, V, and O of CWLP sample surface, (**m**) Microstructure top layer of LPL sample, (**n**) High resolution map, (**o**–**r**) Distribution of Ti, Al, V, and O of LPL sample surface.

**Figure 5 micromachines-15-00336-f005:**
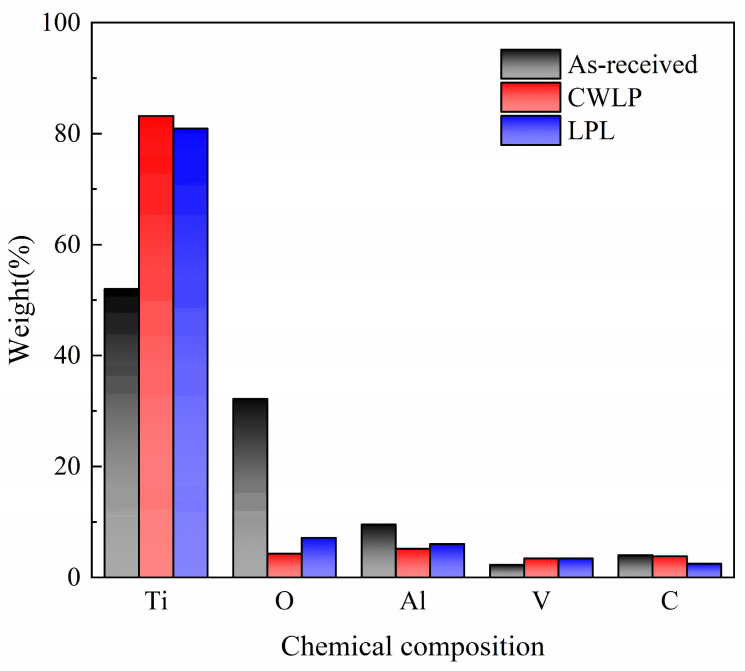
The EDS spectra and the elements content of as-received, CWLP, PLP sample surfaces.

**Figure 6 micromachines-15-00336-f006:**
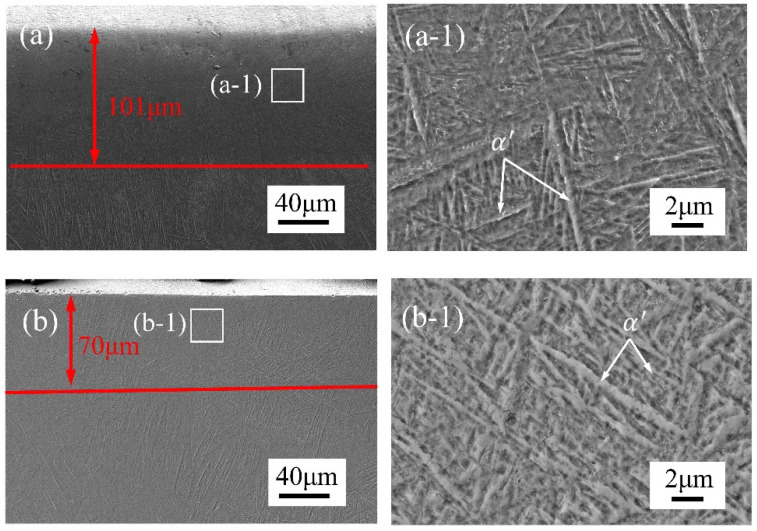
Microstructure in longitudinal section of the SLM Ti_6_Al_4_V alloy samples with different states: (**a**) Microstructure of longitudinal section in CWLP sample, (**a-1**) Higher-magnification SEM images of the CWLP surface, (**b**) Microstructure of longitudinal section in PLP sample, (**b-1**) Higher-magnification SEM images of the PLP surface.

**Figure 7 micromachines-15-00336-f007:**
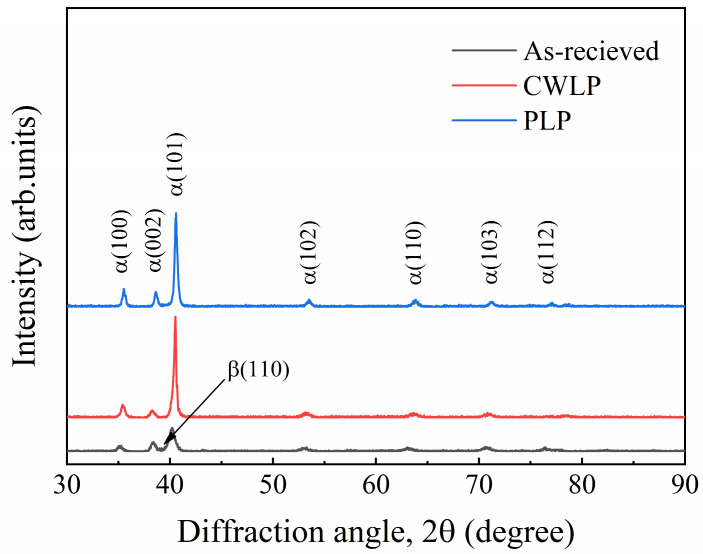
XRD patterns of as-received, CWLP, PLP processed sample.

**Figure 8 micromachines-15-00336-f008:**
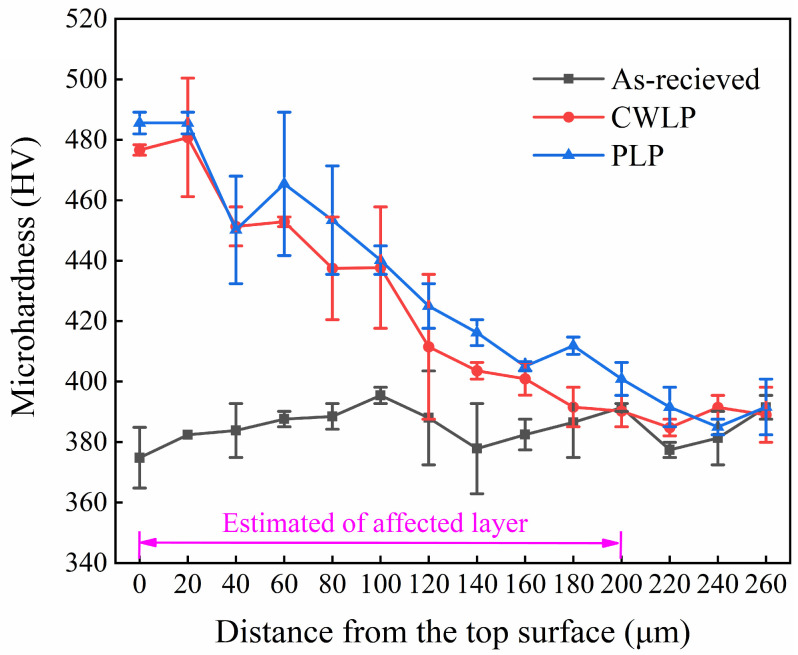
Micro-hardness distribution in cross-section of as-received, CWLP, and PLP samples.

**Figure 9 micromachines-15-00336-f009:**
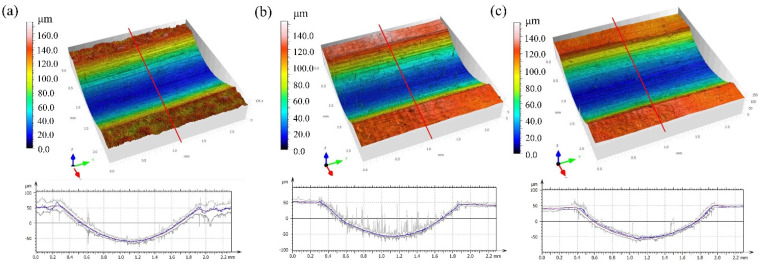
Three-dimensional image of the friction and wear track (**a**) as-received (**b**) CWLP sample (**c**) PLP sample.

**Figure 10 micromachines-15-00336-f010:**
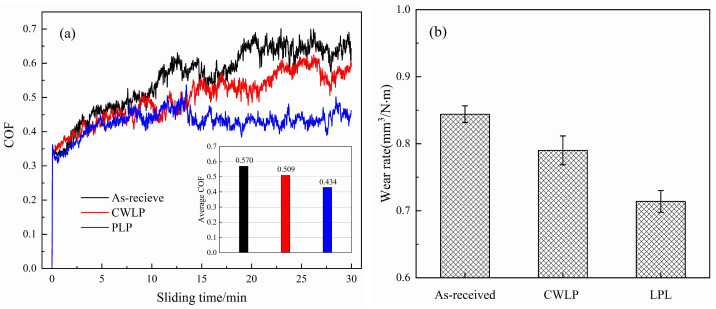
(**a**) Friction coefficient curve and average COF (**b**) wear rate of as-received, CWLP, and LPL samples.

**Figure 11 micromachines-15-00336-f011:**
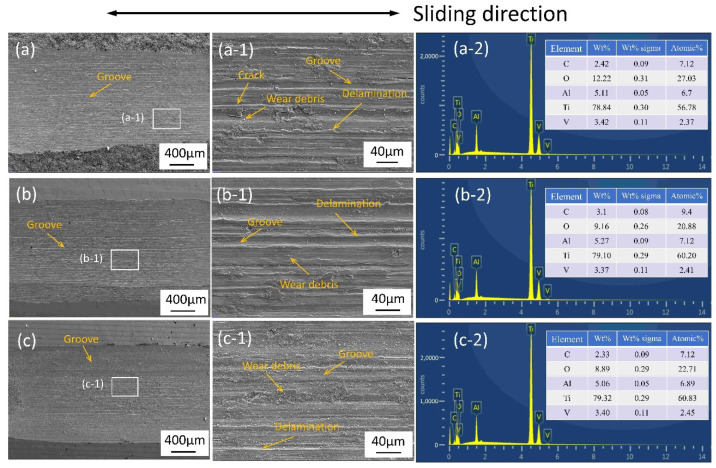
SEM micrographs along with the EDS spectra of the (**a**,**a-1**) as-received sample with different magnification, (**a-2**) EDS result of (**a-1**), (**b**,**b-1**) CWLP sample with different magnification, (**b-2**) EDS result of (**b-1**), (**c**,**c-1**) LPL sample with different magnification, and (**c-2**) EDS result of (**c-1**).

**Table 1 micromachines-15-00336-t001:** Laser polishing parameters used in this experiment.

Process Parameters	P (W)	V (mm/s)	d (mm)	f (KHz)	t (ns)
CW laser polishing	250	70	0.1	-	-
Pulsed laser polishing	120	80	0.08	100	240

## Data Availability

Data are contained within the article.

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
