# Peer review of "Study on Laser Polishing of Ti6Al4V Fabricated by Selective Laser Melting"

_micromachines, 2024, doi:10.3390/mi15030336_

Round 1

Reviewer 1 Report

Comments and Suggestions for Authors

This manuscript brings an important focus on improving surface quality and functional performance of components manufactured by material additive technologies. However I am hesitant recommend this manuscript for acceptance there is a significant number of illogical and unclear statements and explanations. This is the list to begin with:

Title:

-          No abbreviations should be in the title

-          It is illogical “Research on … the surface …”

-          Does not cover all aspects of the performed research

Abstract:

-          “poor surface quality” is not an engineering definition

-          It is not clear how “poor surface quality” is connected with “overall cost and lead time”

-          “lead time” of what?

-          “rapid heating and cooling” is not related to laser polishing because it is “rapid remelting and resolidification”

-          Should be “pulsed laser”

-          Term “surface roughness” is not completely defined; is it profile (Ra) or areal (Sa)?;

-          Ra should not be used for surface quality assessment

-          “decrease slightly” is not an engineering term

-          “dry wear condition” should be replaced on “dry friction condition” as friction is first and wear is second

Keywords:

-          SLM should be spelled out

-          replace “morphology” with “topography”

-          “dry friction” should be added

Introduction:

Page 1:

-          32: “low temperature performance” is not clear

-          35: “more and more … paying more …” stylistically incorrect

Page 2:

-          48: change “reach” to “reaches”

-          51: change “used the fiber” to “used a fiber”

-          55: “et al.” should be in italic font

-          58: who are “they were able”?

-          63: “refinement in grain size” does not sound right

-          65: change “induced” to “induce”

-          66: what does it mean “nano continuous wave polishing”? “nano” does not sound right

-          73: change “fabricate using” to “fabricated using”

-          Entire introduction is not comprehensively written as it appears as a list of facts without their analysis and relation to the current research; it should be completely re-written and include more relative publications (especially publications that show much better results in improving surface quality), such as:

Diaz, A., et al. (2024). Surface post-treatment of additively manufactured components. Additive Manufacturing of High-Performance Metallic Materials: 223-283.

Karlsson, J., et al. (2023 ). "Laser polishing of PBF-LB manufactured stainless steel surfaces." IOP Conf. Series: Materials Science and Engineering 1296 (paper 012023)

Yadav, D. and I. Mingareev (2023). "Utilizing ultrafast lasers for postprocessing to improve mechanical properties of 3D-printed parts." JOURNAL OF LASER APPLICATIONS 35(1)

Chernyshikhin, S. V., et al. (2023). "Laser Polishing of Nickel-Titanium Shape Memory Alloy Produced via Laser Powder Bed Fusion." Metals and Materials International.

Cvijanovic, S. J., et al. (2023). "Applicability of Laser Polishing on Inconel 738 Surfaces Fabricated Through Direct Laser Deposition." Journal of Laser Micro/Nanoengineering 18(1): 7.

Bordatchev, E. V., et al. (2014). "Performance of laser polishing in finishing of metallic surfaces." The International Journal of Advanced Manufacturing Technology 73(1-4): 35-52.

Shipley, H., et al. (2018). "Optimisation of process parameters to address fundamental challenges during selective laser melting of Ti-6Al-4V: A review." International Journal of Machine Tools and Manufacture 128: 1-20.

Comments on the Quality of English Language

Engineering logic and terminology soundness should be significantly increased per indicated above recommendations

Author Response

Micromachines-2883565

Response to Reviewer

Dear Reviewer,

Thank you for giving us the opportunity to submit a revised draft of the manuscript “Study on surface characteristics of Ti6Al4V components by laser polishing” for publication in the Journal of Micromachines. We appreciate the time and effort that you dedicated to providing feedback on our manuscript and are grateful for the insightful comments on and valuable improvements to our paper.

We have incorporated most of the suggestions made by the reviewers. Those changes are highlighted within the manuscript. Please see below, in black, for a point-by-point response to the reviewers’comments and concerns. All page numbers refer to the refer to the revised manuscript file with tracked changes.

  1. No abbreviations should be in the title

Responds: Thank you for your valuable suggestion. I have revised the title of the paper. The revised title is “Study on laser polishing of Ti6Al4V fabricated by selective laser melting” (line 2)

  1. It is illogical “Research on … the surface …”

Responds: Thank you for this suggestion. I have revised the title of the paper. (line 2)

  1. Does not cover all aspects of the performed research

Responds: Thank you for your valuable suggestion. I have revised the title of the paper. (line 2)

  1. “poor surface quality” is not an engineering definition

Responds: Thank you for your valuable suggestion. I have revised it in the manuscript. The revised sentence is “Laser-based additive manufacturing has garnered significant attention as a promising 3D printing method for fabricating metallic components in recent years. However, surface roughness of additive manufactured components has been considered as a challenge to achieve high performance.” (line 10)

  1. It is not clear how “poor surface quality” is connected with “overall cost and lead time”

Responds: Thank you for your valuable suggestion. I have revised it in the manuscript. (line 10)

  1. “lead time” of what?

Responds: Thank you for your valuable suggestion. I have revised it in the manuscript. (line 10)

  1. “rapid heating and cooling” is not related to laser polishing because it is “rapid remelting and resolidification”

Responds: Thank you for your valuable suggestion. I have revised it in the manuscript. (line 16)

  1. Should be “pulsed laser”

Responds: I am really sorry for such errors. I have revised it throughout the manuscript. (line 19)

  1. Term “surface roughness” is not completely defined; is it profile (Ra) or areal (Sa)?;

Responds: Thank you for your valuable suggestion. Surface roughness is quantified as areal roughness (Sa), which is measured using a Laser Confocal Microscope on a 2.5×2.5 mm2 area. I have revised it in the manuscript. (line 20)

  1. Ra should not be used for surface quality assessment

Responds: T Thank you for your valuable suggestion. I have revised it in the manuscript and replaced Ra with Sa for surface quality assessment. (line 20)

  1. “decrease slightly” is not an engineering term

Responds: Thank you for your valuable suggestion. I have revised it in the manuscript. The revised sentence is “Moreover, the coefficient of friction (COF) is slightly reduced by both CWLP and LPL processing, and the wear rate of the surface layer is improved to 0.790 mm3/(N∙m) and 0.714 mm3/(N∙m), respectively, under dry fraction conditions.” (line 25)

  1. “dry wear condition” should be replaced on “dry friction condition” as friction is first and wear is second

Responds: Thank you for your valuable suggestion. I have revised it in the manuscript. (line 27)

  1. SLM should be spelled out

Responds: Thank you for your valuable suggestion. I have revised it in the manuscript. (line 28)

  1. replace “morphology” with “topography”

Responds: Thank you for your valuable suggestion. I replaced surface morphology with surface roughness. (line 28)

  1. “dry friction” should be added

Responds: Thank you for your valuable suggestion. I have revised it in the manuscript. (line 28)

  1. “low temperature performance” is not clear

Responds: Thank you for your valuable suggestion. I deleted the low temperature performance. (line 33)

  1. “more and more … paying more …” stylistically incorrect

Responds: Thank you for your valuable suggestion. I have revised it in the manuscript. The revised sentence is “With the advancement of additive manufacturing technology, an increasing number of researchers are devoting their attention to the utilization of selective laser melting (SLM) technology for the fabrication and processing of titanium alloy products.” (line 35)

  1. change “reach” to “reaches”

Responds: I am really sorry for such errors. I have revised it in the manuscript.

  1. change “used the fiber” to “used a fiber”

Responds: I am really sorry for such errors. I have revised it in the manuscript. (line 54)

  1. “et al.” should be in italic font

Responds: I am really sorry for such errors. I have revised it in the manuscript. (line 54, 58, 61, 65, 68)

  1. who are “they were able”?

Responds: Thank you for your valuable suggestion. I have revised it in the manuscript. The revised sentence is “By controlling the laser power, speed, and beam offset, a significant reduction in surface roughness was achieved, with the initial Ra value of approximately 10.2 μm being reduced to 2.44 μm”. (line 59)

  1. “refinement in grain size” does not sound right

Responds: Thank you for your valuable suggestion. I have revised it in the manuscript. The revised sentence is “Furthermore, the laser polishing induced the phase transformation and a finer grain size.” (line 65)

  1. change “induced” to “induce”

Responds: I am really sorry for such errors. I have revised it in the manuscript.

  1. what does it mean “nano continuous wave polishing”? “nano” does not sound right

Responds: I am really sorry for such errors. I have revised it in the manuscript. (line 68)

  1. change “fabricate using” to “fabricated using”

Responds: I am really sorry for such errors. I have revised it in the manuscript. (line 76)

  1. Entire introduction is not comprehensively written as it appears as a list of facts without their analysis and relation to the current research; it should be completely re-written and include more relative publications (especially publications that show much better results in improving surface quality).

Responds: Thank you for your valuable suggestion. I have made a major modification to the introduction section and added corresponding publication to the manuscript.

Reviewer 2 Report

Comments and Suggestions for Authors

Review comments for

Manuscript ID: micromachines-2883565-peer-review-v1     

Title: Research on Performance of CWLP and PLP on the Surface of Ti6Al4V Components

Shuo Huang, Junyong Zeng, Wenqi Wang and Zhenyu Zhao

Submitted to: Micromachines

Comments:

The submitted manuscript investigated the laser-induced polishing method of Ti6Al4V metal alloy composite material. For the laser-induced polishing method, the authors used continuous lasers as well as pulsed lasers for the investigation.

The manuscript is planned and organized well. All the experiments are well organized giving a clear picture of the conclusions to the readers. However, the manuscript is poorly written. Figures are reasonably good and acceptable however, the manuscript text has been poorly written having numerous errors.

Having said that the manuscript is poorly written and needs major improvements that need to be addressed seriously before a publication version can be recommended. The current version of the manuscript should be recommended as a “Major Revision”. If the authors make a sincere effort to improve the manuscript texts and scientific clarity for future readers, I will be happy to see the modified revised version before publication.

The issues listed below are critical for the publication and need major consideration to justify all manuscript claims, logical justifications, and clarity.

1)      There must be a gap (empty space) between numerical values and their units. For example, it should be “50 μm” instead of “50μm”. This error is present throughout the manuscript and must be corrected in the revised manuscript version.

2)      Similarly, there should be a gap (empty space) before and after a “=” symbol. For example, see Fig. 3a,b,c. It should be “Sa = 6.29 μm” instead of “Sa=6.29μm” as written in Fig. 3a. This error is represented throughout the manuscript and should be corrected in the revised version carefully.

3)      The number of Atoms in the compound nomenclature should be written in subscript format. For example, it should be Ti6Al4V instead of Ti6Al4V. This error is present throughout the manuscript as well and should be corrected throughout the revised manuscript.

4)      Fig. 3: It is clear from Fig. 3 that, the average surface roughness (Sa) values are reduced in PLP (Sa = 890 nm), while CWLP is a slightly higher Sa value (Sa = 940 nm) in contrast to as received Sa value of around (Sa = 6.29 μm). This is good however only by looking at the three-dimensional morphology plots, it is confusing to the readers, and seems that PLP (Fig. 3c) has a higher rough surface (color trench and valley profile) than as received sample (Fig. 3a). This has happened because the all the plots have been plotted with different color scale bars. Authors are suggested to plot all three plots again with unchanged/fixed color scale bar values. This can be added to the supporting information figures if authors do not find it suitable to add to the main manuscript figures. But this new figure is important to the readers to demonstrate clearly that PLP has the lowest 3D morphology. Currently, this is confusing to the reader and should be corrected.

5)      Fig. 6: Authors mentioned on Page 5, line 188 “ the remelting layer of PLP samples exhibits a higher proportion of ɑ' phase martensite compared to that of CWLP samples.”. However, why PLP samples exhibited a higher proportion of α' phase martensite, is not described and is confusing to the readers. A detailed explanation should be added in the revised manuscript describing the reasons behind this observation.

6)      On page 5 line 181, the author wrote “α' phase martensite” and on line 186 the author wrote “ɑ' phase martensite”. The symbol “α” in both cases is different. Authors are suggested to maintain a single symbol for α throughout the revised version of the manuscript.

7)      The same error as above was found in Fig. 7 description on page 7. This should be corrected as well.

Comments on the Quality of English Language

English is fine with minor errors here and there. However, the manuscript's scientific scripts are poorly written with a lot of error throughout the manuscript. This must be corrected in the revised manuscript.

Author Response

Micromachines-2883565

Response to Reviewer

Dear Reviewer,

Thank you for giving us the opportunity to submit a revised draft of the manuscript “Study on surface characteristics of Ti6Al4V components by laser polishing” for publication in the Journal of Micromachines. We appreciate the time and effort that you dedicated to providing feedback on our manuscript and are grateful for the insightful comments on and valuable improvements to our paper.

We have incorporated most of the suggestions made by the reviewers. Those changes are highlighted within the manuscript. Please see below, in black, for a point-by-point response to the reviewers’comments and concerns. All page numbers refer to the refer to the revised manuscript file with tracked changes.

  1. There must be a gap (empty space) between numerical values and their units. For example, it should be “50 μm” instead of “50μm”. This error is present throughout the manuscript and must be corrected in the revised manuscript version.

Responds: I am really sorry for such errors. I have revised it throughout the manuscript. (line14, 21, 27)

  1. Similarly, there should be a gap (empty space) before and after a “=” symbol. For example, see Fig. 3a,b,c. It should be “Sa = 6.29 μm” instead of “Sa=6.29μm” as written in Fig. 3a. This error is represented throughout the manuscript and should be corrected in the revised version carefully.

Responds: I am really sorry for such errors. I have revised it throughout the manuscript. (line142)

  1. The number of Atoms in the compound nomenclature should be written in subscript format. For example, it should be Ti6Al4V instead of Ti6Al4V. This error is present throughout the manuscript as well and should be corrected throughout the revised manuscript.

Responds: I am really sorry for such errors. I have revised it throughout the manuscript. (line2, 22, 25)

  1. Fig. 3: It is clear from Fig. 3 that, the average surface roughness (Sa) values are reduced in PLP (Sa = 890 nm), while CWLP is a slightly higher Sa value (Sa = 940 nm) in contrast to as received Sa value of around (Sa = 6.29 μm). This is good however only by looking at the three-dimensional morphology plots, it is confusing to the readers, and seems that PLP (Fig. 3c) has a higher rough surface (color trench and valley profile) than as received sample (Fig. 3a). This has happened because the all the plots have been plotted with different color scale bars. Authors are suggested to plot all three plots again with unchanged/fixed color scale bar values. This can be added to the supporting information figures if authors do not find it suitable to add to the main manuscript figures. But this new figure is important to the readers to demonstrate clearly that PLP has the lowest 3D morphology. Currently, this is confusing to the reader and should be corrected.

Responds: Thank you for your valuable suggestion. I re-observed the three-dimensional morphology of CWLP and LPL samples by laser confocal microscopy, subsequently incorporating these findings into the paper. Additionally, I added a bar chart with corresponding errors in Fig. 3. (line 142)

  1. Fig. 6: Authors mentioned on Page 5, line 188 “the remelting layer of PLP samples exhibits a higher proportion of ɑ' phase martensite compared to that of CWLP samples.”. However, why PLP samples exhibited a higher proportion of α' phase martensite, is not described and is confusing to the readers. A detailed explanation should be added in the revised manuscript describing the reasons behind this observation.

Responds: Thank you for your valuable suggestion. I have added the detail explanation in the revised manuscript. “In contrast to CWLP that have a steady energy input, pulsed lasers exhibit intermittent energy input, resulting in a higher cooling rate and an increased formation of martensitic structure on the surface of laser-polished samples. According to the study of Ma et al. [16], in thermal cycles of pulsed laser processing, the cooling rates of Ti6Al4V was calculated 1.209 × 104 K/s, which are much higher than 410 K/s. Due to the rapid solidification and cooling rates, the bcc β phase undergoes a complete transformation into the metastable hcp α′ phase martensite phase through a diffusionless and shear-type transformation process. Therefore, the remelting layer of PLP sample exhibits a higher proportion of α′ phase martensitic structure”. (line 194)

  1. On page 5 line 181, the author wrote “α' phase martensite” and on line 186 the author wrote “ɑ' phase martensite”. The symbol “α” in both cases is different. Authors are suggested to maintain a single symbol for α throughout the revised version of the manuscript.

Responds: I am really sorry for such errors. I have revised it throughout the manuscript. (line 187, 192)

  1. The same error as above was found in Fig. 7 description on page 7. This should be corrected as well.

Responds: I am really sorry for such errors. I have revised it throughout the manuscript. (line 223, 226, 229)

Reviewer 3 Report

Comments and Suggestions for Authors

This paper describes the effect of two laser polishing techniques on the  surface of the Ti6Al4V alloy. The paper has a lot of experimental information about the study question, but there is still missing important information to assess the final results of the two polishing techniques:

- The most suitable technique to assess the surface roughness of the material is AFM. This technique should be used to obtain the experimental surface roughness.

- A statistical analysis of the experimental results about the roughness must be done using significant experimental measurements with the corresponding errors. The comparison must be supported by the required statistical tests analysis.

- A statistical analysis should be done for the other characteristics of the polishing materials surface. 

Author Response

Micromachines-2883565

Response to Reviewer

Dear Reviewer,

Thank you for giving us the opportunity to submit a revised draft of the manuscript “Study on surface characteristics of Ti6Al4V components by laser polishing” for publication in the Journal of Micromachines. We appreciate the time and effort that you dedicated to providing feedback on our manuscript and are grateful for the insightful comments on and valuable improvements to our paper.

We have incorporated most of the suggestions made by the reviewers. Those changes are highlighted within the manuscript. Please see below, in black, for a point-by-point response to the reviewers’comments and concerns. All page numbers refer to the refer to the revised manuscript file with tracked changes.

  1. The most suitable technique to assess the surface roughness of the material is AFM. This technique should be used to obtain the experimental surface roughness.

Responds: Thank you for your valuable suggestion. However, due to the limitation of experimental conditions, I could not assess the surface roughness of the materials by AFM.

  1. A statistical analysis of the experimental results about the roughness must be done using significant experimental measurements with the corresponding errors. The comparison must be supported by the required statistical tests analysis.

Responds: Thank you for your valuable suggestion. I added a bar chart with corresponding errors in Fig. 3. Additionally, I also added corresponding statistical analysis in the manuscript. (line 134, 142)

  1. A statistical analysis should be done for the other characteristics of the polishing materials surface.

Responds: Thank you for your valuable suggestion. I added corresponding statistical analysis in the manuscript. (line 276)

Round 2

Reviewer 2 Report

Comments and Suggestions for Authors

Review comments for

Manuscript ID: micromachines-2883565-peer-review-v2     

Title: Study on laser polishing of Ti6Al4V fabricated by selective laser melting

Shuo Huang, Junyong Zeng, Wenqi Wang and Zhenyu Zhao

Submitted to: Micromachines

Comments:

The authors have modified the manuscript accordingly. The modifications improved the manuscript's quality and readability. I recommend the manuscript for publication.

Reviewer 3 Report

Comments and Suggestions for Authors

The authors have answered satisfactorily to the refere comments.